# Age-Related Changes in Survival Behaviour in Parasite-Free Hatchery-Reared Rainbow Trout (*Oncorhynchus mykiss*)

**DOI:** 10.3390/ani14091315

**Published:** 2024-04-27

**Authors:** Rafael Freire, Mathea Michie, Leia Rogers, Shokoofeh Shamsi

**Affiliations:** Gulbali Institute, Charles Sturt University, Elizabeth Mitchell Drive, Albury, NSW 2640, Australialrogers@csu.edu.au (L.R.); sshamsi@csu.edu.au (S.S.)

**Keywords:** fish, behaviour, welfare, reintroduction, restocking

## Abstract

**Simple Summary:**

Hatchery-reared Rainbow trout are released into freshwater systems, often to support recreational fishing, but the best age to release fish to maximise their chances of survival is unknown. Using laboratory tests of fish behaviour at six ages, we report age-related changes in emergence, exploration, food neophobia, habitat choice and predator avoidance. We suggest that an understanding of age-related changes in behaviour should be applied to support management decisions around the best age to release Rainbow trout.

**Abstract:**

Millions of hatchery-reared Rainbow trout are currently released in Australian waters to support recreational fisheries objectives, yet many of these fish die soon after release. In addition, little is known whether these fish harbour parasites that can potentially threaten freshwater ecosystems and human health. Here, we tested the behaviour of hatchery-reared trout using six tank-based tests at six different ages to evaluate their chances of survival and then dissected fish to investigate parasite prevalence. At 7 weeks of age fish readily emerged from a hide and showed the greatest number of startle responses to predators. Behaviour around 25–29 weeks of age was relatively “shy”, staying in shelter and avoiding open water. At around 37–41 weeks of age though, behaviour changed, with fish emerging from a hide more readily and exploring the environment. Interestingly, at 58 weeks of age fish were slower to initiate exploration, possibly indicating a return to “shyer” behaviour. All fish underwent thorough parasite examination, revealing no infections. We conclude that knowledge of the behaviour of hatchery-reared fish at different ages is useful for decisions around the timing of release that balance the needs of recreational fishers whilst managing the impact on freshwater ecosystem.

## 1. Introduction

Millions of hatchery-reared fish are released to freshwater systems to support recreational and conservation objectives, yet it has been known for some time that most of these fish do not survive. A major cause of mortalities of these fish in the wild is due to deficiencies in behavioural development arising from an inadequate captive rearing environment [1]. In particular, captive fish that are confined in a simplified and predator-free environment develop behavioural deficiencies such as a failure to recognise and respond to fish predators [2]. The outcome is that during rearing in hatcheries juvenile fish can become adapted to captivity conditions through environmentally-induced developmental adaptations [3]. In addition, many freshwater fish populations face significant population pressure from a range of factors, such as habitat degradation, altered hydrology, invasive species and climate change [4]. The behaviour of fish is the first line of defence in response to these anthropogenic challenges and the type and timing of behavioural responses will determine whether fish survive, and ultimately explain population dynamics [5]. An understanding of hatchery-reared fish behaviour at the time of release is therefore crucial to evaluate their chances of survival after release.

As well as phenotypic changes brought about by the rearing conditions, the age of the individual at the time of release can have an impact on the success of the release [6]. Fish that are very young and small at the time of release may be vulnerable to more predators and may not have developed the appropriate behaviour for survival such as food acquisition. One solution is to release bigger fish; for example, in Australia, hatcheries release bigger Rainbow trout (*Oncorhynchus mykiss*) that are better able to mitigate the negative effects of predation [7]. It should also be noted that keeping fish to an older age will incur a higher economic cost on hatcheries but, as noted above, the drawback is that if individuals are held in captivity for too long, they become overly habituated to captivity [6,8]. Captive-reared animals generally show increased boldness and risk-taking behaviour, often described as a “domestic” phenotype (though note that this may, or may not, be in conjunction with artificial selection) [5]. Although older individuals may be more habituated to captivity, studies on a range of animals, including fish and birds, have shown that release programs are more successful when the individuals released are at an older age [6,9]. This may be because their bolder behaviour, a common result of living in captivity, allows them to be more efficient foragers and be less fearful of moving around the environment for food and other resources [10,11].

New South Wales (NSW) and Victoria release approximately 2.5 million Rainbow trout annually [12]. Trout are the predominant fish of choice for anglers and their presence in popular fishing waterways increases tourism revenue as well as drawing in profits from fishing licenses, with recreational fishing as a whole contributing to 100,000 jobs and $11 billion to the Australian economy each year [13]. Although there are many economic benefits to recreational fishing, supported by the release of Rainbow trout, the continuation of this practice brings controversy and has been regarded as contrary to conservation efforts [14,15]. Much of the concern around the introduction of exotic fish species centres on the significant disruptions to the native communities through direct and indirect detrimental impacts.

Ecological research into the impacts of trout on Australian ecology is increasing and is making the impacts of trout on the environment and native fauna more recognised [16]. Trout pose a threat to native fish through direct predation and out-competing native species [14,17] which can lead to population fragmentations and disrupt the ranges occupied and available to native species [15,18]. Trout are a predatory species that feed on aquatic fauna such as small fish and crustaceans, and have been known to eat frogs [12]. For example, native Australian species such as the endangered stocky galaxiid (*Galaxias tantangara*) are highly affected by trout predation to the point of local extinction in some areas of NSW [12,18]. Trout research focuses on either the impacts of their release on the environment or the management strategies of hatchery trout. However, there is limited interaction between these two fields, hindering our understanding of how one may affect the other [6]. By studying the behaviour of hatchery-reared trout, we can gain a better understanding of their interaction with the environment after release and better predict their impact on the ecosystem.

In addition, released fish may pose a threat to the health of other freshwater animals, and human fish consumers, through the accumulation and spread of parasites [19]. The intensive environment in which trout are reared can create conditions highly favourable for parasite breeding and infections [20], especially when there is poor hatchery management. Fish can then be overloaded with parasites at the time of release and increase the prevalence of certain parasites at release sites [21,22]. In other parts of the world, a range of parasites including protozoa and metazoa have been reported in hatchery-reared Rainbow trout [23,24], though little is known about the parasites of Rainbow trout in Australia. Understanding the threat that fish parasites pose to humans is also critical for ensuring food safety, and is a major focus of the World Health Organisation with so much of the world’s population relying on freshwater fish for food [25]. Unfortunately, there is minimal evidence of the parasite prevalence within hatchery systems which can be detrimental to whole ecosystems and humans [22].

Lastly, parasitic organisms are also capable of altering the behaviour of their hosts. This can be a strategic adaptation that facilitates the parasites’ movement along the food web, or the parasites can alter behaviour by making fish sick, or by imposing a metabolic demand on the fish that changes foraging and other resource-acquiring behaviour [20]. Despite increasing evidence on the impact of parasites on fish behaviour, little consideration has been paid to the ecological implications of the influence of parasites on fish behaviour [26]. By deepening our understanding of trout behaviour and health, we can improve our knowledge of the part parasites play in influencing behaviour and the wider ecosystem they inhabit.

Here, we tested the behaviour of hatchery-reared Rainbow trout in six tests at six different ages to evaluate their chances of survival and impact on the ecosystem. Behaviour was studied using a bold–shy framework [5], where boldness is a measure of an individual’s tendency to take risks and encapsulates how an individual responds to risk and novelty. Bolder animals are generally more active and more likely to explore novel objects or environments and spend more time away from shelter [11]. Our behavioural tests were designed to investigate four main facets of boldness behaviour that are crucial for the survival of fish in the wild: the ability to (1) find and use shelter, (2) explore and find food and other resources, (3) avoid predation and (4) find and exploit appropriate habitat. We predicted that in response to captive rearing, older fish would be bolder as revealed by less time in shelter, more exploration, taking more risks with predators and more readily taking novel foods. We also predicted that older fish would have more parasites than younger fish.

## 2. Materials and Methods

### 2.1. Subjects

A total of 125 Juvenile Rainbow trout (*Oncorhynchus mykiss*) from Gaden Trout Hatchery (NSW) were used (Table 1). Fertilised eggs were cultured with a flow-through system of water from the Thredbo river. Eggs and fry were hatched and raised in larval troughs and tanks until fingerlings were moved and reared on commercial fish food in outdoor hatchery ponds, raceways and tanks.

Fish were transported to CSU’s Freshwater laboratory and kept in a recirculating tank system (total volume of 1000 L) with a biological filter to manage nitrous waste and UV treatment to control infectious agents. The system was kept at 11–12 °C, a dissolved oxygen level of 9–10 mg/L and a pH range between 6.5 to 7.5. Testing kits for ammonia, nitrite, nitrate and alkalinity (API freshwater testing kits, Mars Inc., Chicago, IL, USA) were used to monitor water parameters every second day. Other water parameters were monitored with a Horiba U-52 (Horiba Advance Techno Co., Kyoto, Japan) multi-parameter water quality meter. Behavioural tests were undertaken in two separate tanks that were connected to provide the same water as the holding aquariums and a summary of all behavioural tests is provided in the Appendix A. All fish received all six tests in a fixed order designed to minimise fish handling.

### 2.2. Behavioural Tests

#### 2.2.1. Emergence, Exploration, Habitat Choice and Bird Predator Tests

Four behavioural tests were performed sequentially in a Plus maze (Figure 1 and Appendix A), in the order of emergence, exploration, habitat choice and response to bird predators, to minimise fish handling. Video cameras (Dahua 5231 Startlight, Dahua Technology Pty Ltd., Artarmon, Australia) were fitted above the testing tanks to record behaviour from outside the testing room.

The emergence test involved the fish being placed into the refuge (marked R, Figure 1). After 10 min, a guillotine door was raised 10cm allowing the fish to exit the refuge into A1. The latency for the whole fish to emerge from the refuge was recorded. A cut-off period of 10 min was used if the fish did not emerge. The fish was then allowed to swim around the entire Plus maze and the latency to exit Arm 1 (referred to as initiating exploration) and the total number of arms of the Plus maze that the fish entered was recorded in 10 min. Following the exploration test, the fish was confined in the centre of the Plus maze and four substrates (gravel, pebbles, plants or no substrate) were placed in each arm of the Plus maze (Appendix A). A pseudo-random sequence was used to select the position of each substrate. The fish was allowed to acclimatise in the centre for 5 min and then released and its location was recorded every minute for 10 min.

Following the habitat choice test, the fish was confined at one end of the plus maze for 5 min near PT (Figure 1), but with a divider preventing view of the model bird stimulus. The model bird was a taxidermised Nankeen Night Heron (*Nycticorax caledonicus*, Appendix A), placed on a turn table rotating through 90° every 4 s. The dividers were then removed which allowed the fish to swim away from the bird predator. The response, either avoidance, freeze (no movement of fins) or no response, and latency to move away from the predator was recorded for 10 min.

#### 2.2.2. Fish Predator Test

The following day, the Rainbow trout was placed in a rectangular tank (80 × 30 cm) on the other side of the opaque barrier to a 25 cm Murray cod (Appendix A). After 10 min, the opaque barrier was removed, revealing the perforated, transparent barrier that physically separated the Murray cod from the test fish. The same behavioural variables were recorded as in the model bird predator test and the test concluded after 10 min.

#### 2.2.3. Novel Food Test

The fish was then moved back to its individual tank for at least a two-hour acclimation period, prior to receiving a novel food test. A Gulp! Alive non-edible floating salmon egg (Berkley, Spirit Lake, IA, USA) was placed on a blunted, #10 barbless hook (Owner, SBL-55 M, Owner Hooks, Costa Mesa, CA, USA). The hook and bait were attached to a force-displacement transducer (Imada DST-50, Imada, Northbrook, IL, USA). With the use of the force transducer and the security cameras, a bite was confirmed as a reading greater than 0.01 N and was confirmed by an observer looking at the live feed in an adjoining video room. The time to the first confirmed bite was recorded to the nearest minute, with a maximum of 60 min.

### 2.3. Study of Parasites

Fish were euthanised at the end of all behavioural tests by Aqui-S (Huber group, Lower Hutt, New Zealand) overdose at a ratio of 3 mL:10 L water and a subsequent 1:1 ice slurry. The fish were then examined for parasitic infections according to standard protocols [27]. The gills were removed, and the individual gill filaments examined under a dissecting microscope. The fish were opened from the anus to between the pelvic fins and the internal organs removed for examination; all internal organs including the gut, gonads, swim bladder, liver, kidney, pyloric area and heart were thoroughly examined for the presence of parasites. The body wall was removed by making incisions above the pectoral fins and along the lateral lines to the anus. The brain was removed and a sample was placed on a microscope slide for inspection under a compound microscope. The jaw and eyes of the fish were also examined. All procedures were approved by Charles Sturt University’s Animal Ethics Committee (A22338).

### 2.4. Statistical Analysis

Statistical analysis was conducted using the statistical package R [28]. The response variables latency to emerge, in the emergence test, and latency to bite the novel food were heavily skewed and were converted to binary variables (emerged/did not emerge and bit the bait/did not bite the bait, respectively). Emergence and biting the bait were analysed using Pearson’s chi-squared test, fitted against age and a chi-square post hoc test with a Bonferroni adjustment used for further analysis of a significant result. Time spent in each habitat in the habitat preference test was analysed using Pearson’s chi-square test on a two-way contingency table of age by habitat type with a chi-square post-hoc test with a Bonferroni adjustment used for further analysis.

The response variables latency to exit, arms entered and the number of startles for avian and fish predators were not normally distributed and were analysed using the non-parametric Kruskal–Wallis rank sum test with the response variable fitted against age. A pairwise Wilcox test with a continuity correction was used as the post hoc test for a significant Kruskal–Wallis result.

For a comparison between the trout’s response to a bird predator and a live fish predator, the response variables were fitted to a generalised linear mixed model (GLMM) with age and predator type as factors, and fish identity as the random variable. For the variable response type (freeze or move away), a binomial logistic GLMM was fitted and for latency to show an anti-predator response and the distance from the predator a Gaussian identity GLMM was fitted. Multiple comparison pairwise post hoc tests were conducted with a Tukey’s adjustment.

## 3. Results

### 3.1. Behaviour Tests

#### 3.1.1. Emergence and Exploration Tests

Emergence was significantly influenced by age (χ^2^ = 54.17, *p* < 0.0001; Figure 2). Chi-squared post-hoc pairwise analysis indicated that 7-week-old trout all emerged from refuge within 10 min (Table 2). In contrast, at 25 and 29 weeks old, trout were significantly more likely to stay in refuge than expected (All *p* < 0.01; Table 2). Behaviour changed again with 37-weeks-old trout significantly more likely to emerge than expected (*p* = 0.02; Table 2). At 41 and 58 weeks of age, trout were equally likely to emerge and not emerge (NS). A boxplot of latency to emerge is shown in Appendix A.

The latency for trout to exit arm 1 in the exploration test, signifying the start of exploration, was significantly influenced by age (χ^2^ = 21.32, *p* < 0.001; Figure 3a). Dunn’s post-hoc tests indicated that 58-week-old trout were significantly slower to initiate exploration than trout at other ages (All *p* < 0.001). Interestingly, once the trout exited arm 1, they showed significant differences in levels of explorations (χ^2^ = 31.68, *p* < 0.0001; Figure 3b). Pairwise post-hoc tests indicated that in general 37-week-old trout entered significantly more arms than at other ages (Dunn’s post-hoc tests, Appendix A).

#### 3.1.2. Habitat Choice Test

There were significant differences in habitat preferences between the six ages (χ^2^ = 149.93, *p* < 0.0001; Figure 4). Post-hoc tests indicated that 29-week-old trout spent significantly less time in the empty arm (residual −6.6, *p* <0.0001), 37-week-old trout the least time in the plant arm (residual 5.1, *p* < 0.0001) and 58-week-old trout spent significantly more time in the empty arm (residual 5.8, *p* < 0.0001).

#### 3.1.3. Predator Response

Trout moved away from the predators soon after exposure on 82% of occasions, with 18% of the time fish showing an immediate freeze response. A significant predator/age effect was found in the latency to move away from the predator (χ^2^ = 29.16, *p* < 0.0001; Figure 5a). This was attributed to 58-week-old trout being significantly slower to move away from the bird than trout at other ages (estimates, F = 255, df = 119, *p* < 0.0001). A significant predator/age interaction effect was also found in the number of startle responses (χ^2^ = 65.06, *p* < 0.0001; Figure 5b). This was due to 7-week-old fish showing the greatest number of startle responses to the bird predator (Figure 5b). The type of response (moving away or freezing) was not influenced by predator type (χ^2^ = 0.006, *p* = 0.93), age (χ^2^ = 8.00, *p* = 0.16) or predator/age interaction (χ^2^ = 1.58, *p* = 0.90).

#### 3.1.4. Novel Food Test

Overall, 43.2% of trout bit the bait within one hour of presentation. Age had no significant effect on the percentage of trout biting the bait (χ^2^ = 10.8, *p* = 0.054; Appendix A). A correlation coefficient of r = 0.26 suggests that trout were reluctant to bite the bait at 7 and 25 weeks of age compared to older ages. Latency to bite the bait was also not influenced by age (χ^2^ = 10.45, *p* = 0.07).

### 3.2. Parasite Prevalence and Other Findings of Fish Dissections

All fish were found to be in good health with a 0% parasite prevalence observed in 125 subjects. The flesh of the fish was a healthy white with no notable discolouration. The organs in the body cavity appeared to be healthy with no notable signs of discolouration or internal damage; the livers and heart were a healthy red and the gastrointestinal tract appeared to be in good condition with no signs of previous infections (e.g., Appendix A). The gill filaments of the fish dissected were a healthy pink colour with structured vasculature and blood supply to the fingers. There were no obvious signs of previous infections, such as scarring, lesions or organ damage, within these fish so it could be assumed that there has been no recent exposure to parasites.

## 4. Discussion

The youngest age that we tested, 7 weeks of age, showed fast emergence from a hide and the most startle responses in our predator tests. Andersson et al. [29] similarly found that young Rainbow trout readily emerge from a hide and linked this behaviour to the need to establish a territory and out-compete other fish for resources. The fast-emergence behaviour around this age also presents in fish with an increased risk of predation, and this was perhaps compensated by the significantly greater number of startle responses to our predators at this age. The behavioural responses of fish at 7 weeks of age in our tests were largely in line with what would be expected to be natural behaviour, but it would be useful to conduct behavioural tests on wild-caught fish at this age to confirm this. In addition, and this was a limitation of our tank-based study, further tests under different environmental conditions, such as moving water at different velocities and different levels of turbidity, would be useful to confirm if all the behavioural changes reported here are seen under varied natural conditions. If confirmed, releasing fish at a younger age may be the best option for ensuring they pose a suitable behavioural phenotype for survival, but this should be weighed against the increased risk of predation that young fish may experience.

At the next two ages that we tested, 25 and 29 weeks of age, behaviour was also largely in line with the natural behaviour of Rainbow trout. At these ages fish readily used the hide for protection, showed moderate levels of exploration and avoided open water, predators and novel foods. Trout in the wild can be subject to strong water flow and challenging environments [30], and prey will tend to come to hiding fish [31], conditions that would favour the relatively “shy” behaviour seen in our tests at these ages. Combined, these findings suggest that fish at these ages were relatively shy, which would appear to be an appropriate behavioural strategy for this size fish to maximise their chances of survival.

At 37 and 42 weeks of age, however, behaviour changed considerably, and fish showed bolder behavioural traits than at younger ages. At these ages, they were more likely to emerge from a hide, showed significantly more exploration and tended to be more likely to bite the novel food than at other ages. Boldness and shyness have been known for some time to change in the same individual in response to environmental stimuli. Rainbow trout, for example, have been shown to change their boldness in response to the behaviour of conspecifics and show variation in boldness linked to growth and context [32,33]. In addition, captive-bred brown trout (*Salmo trutta*) are faster to explore an environment in a context where a known food source is placed in the experiment arena [32]. Rainbow trout have been previously considered to be a species where increased boldness and the expression of explorative behaviours are useful for defending or expanding their territory and would therefore be a beneficial behavioural adaptation [34]. Although the shift towards bolder behaviour at 37–41 weeks of age may be a result of fish adapting to the captive environment, further tests with wild-caught fish at these ages are necessary to confirm if these behavioural changes are caused by the rearing environment. Nonetheless, when considering the implications of these findings for release practices, it would be important to note that these fish are likely to exhibit boldness when released, and may compete with native fish for food and other resources.

It was interesting to note that the anti-predator behaviour of our trout was largely not influenced by age. In general, all trout tested showed appropriate anti-predator behaviour, even though they had never encountered a predator. Apart from the high number of startle responses seen in 7-week-old fish discussed earlier, the only other finding of note was that 58-week-old fish were slower to move away from a bird predator than fish at other ages. This latter finding, however, could be due to 58-week-old fish being too big to be threatened by our model bird predator. Captive-bred trout are inherently predator naïve and as such were expected to exhibit a reduced predator response. Our findings suggest that anti-predator behaviour in Rainbow trout is largely innate, although in fish it is also likely to have a learned component. Giles and Huntingford [35] found that differences in observed anti-predator responses in the three-spined stickleback (*Gasterosteus aculeatus*) related to the estimated predation risk at study sites with fish from areas of high predation risk having generally higher response rates compared to more predator-naïve fish, showing that while anti-predator behaviour may be inherited, these behaviours are also influenced by experience.

It is important to note that hatchery selection can reduce anti-predator behaviour [36], but this is not the case for our fish since they were F1 progeny of wild genotypes. Johnsson et al. [36] also found that captive-bred brown trout exhibit less selection against risky behaviours and excessive aggression, leading to a higher competitive ability and a less pronounced predator response. The findings of Johnsson et al. [36] determined that due to the altered behavioural patterns observed in captive-bred brown trout, they may pose a threat to wild populations. Brown trout are a more successful invasive species compared to Rainbow trout, though Rainbow trout have higher levels of aggression in novel situations [37]. When considering the impact of released trout on ecosystems, the above findings suggest that breeding from wild-caught adults may be beneficial as these fish may be shyer and have less impact on the ecosystem.

Interestingly the changes towards a “bolder” phenotype at around 37–41 weeks of age appear to reverse somewhat, with 58-week-old fish being less likely to emerge from a hide and showing a reduction in exploration at this age. Polverino et al. [38] reported similar changes towards a “shyer” behavioural profile at older ages in Eastern mosquito fish (*Gambusia holbrooki*), which may be related to a greater ability to inhibit impulsive behaviour at older ages in these fish.

Our Rainbow trout showed little neophobia towards a baited hook suggesting a high probability of being hooked by anglers. Using a similar test, Freire et al. [39] found that captive-bred Spangled perch (*Leipotherapon unicolor*), a successful invasive species, took on average over an hour to respond to a novel baited hook. Neophobia is only one factor that could influence catch rates of fish with other external and internal factors contributing to this. Lennox et al. [40] conceptualise this catch rate as a dynamic state that incorporates the internal state of the fish, the use of different fishing gear and encounters with the ‘predator’ (the fisher). Our findings suggest that release of captive-bred trout, in this respect, appears to support the expectations of recreational anglers.

The presence of parasites, particularly those with the ability to manipulate host behaviour, is a well-documented aspect of trout ecology [23]. Several parasites utilise trout as their intermediate hosts, creating a dynamic relationship that can influence the behaviour of these fish. Interestingly, certain parasites, like *Eustrongylides* larvae, have been identified as common parasites of trout in Australian freshwater systems [41]. However, we found no parasites in our fish and therefore no parasitic influences on the behaviour exhibited by the subjects. The absence of parasites in the sample subjects is most likely due to the water quality at the Gaden Hatchery, though treatment and prevention practices can also significantly reduce the incidence of disease and parasite infection. Parasite management often relies on chemical therapy immersions, generally of either sodium percarbonate or formalin [21,23]. Water temperature and the chemical therapies used influence the prevalence of parasites and the types of parasites found, with some parasites, such as diplomonad species, showing higher abundances at temperatures between 1 and 5 °C, whereas ciliophoran species prefer warmer water temperatures of 16 to 20 °C [23]. Another consideration for hatchery management is the water source used throughout the hatchery, with flow-through systems that use a surrounding water source, harder to manage in terms of regulating water temperature and the infectious agents that come through the system [23].

## 5. Conclusions

Our findings suggest that around 37–41 weeks of age Rainbow trout show a shift towards “bolder” behaviour that may be a result of the captive environment. Before this age behaviour is likely to be quite similar to the behaviour of wild trout. Interestingly though, it appears that there are age-related changes in behaviour that by 58 weeks of age, fish are shyer than at 37 and 41 weeks of age, and so may be better at avoiding predators and surviving in the wild. It is important to note that our study was undertaken in clear, non-moving water, and further tests should be undertaken to evaluate how water velocity and environmental conditions impact fish behaviour. Our examination of fish for parasites strongly supports the current practices at Gaden Hatchery in promoting excellent fish health. Regardless of the contention surrounding the release of exotic fish for recreational purposes, understanding how fish acquire survival behaviour in a captive rearing environment and the potential spread of parasites through these systems and into the wild can help us develop better captive breeding programs, for both recreation and conservation purposes.

## Figures and Tables

**Figure 1 animals-14-01315-f001:**
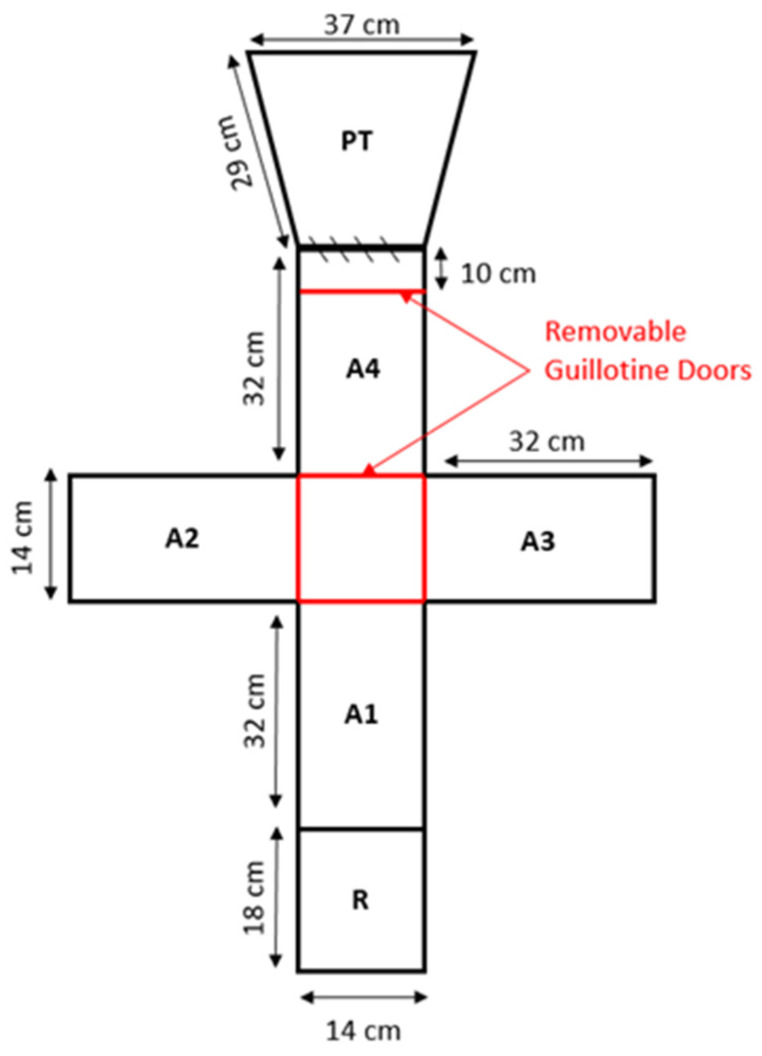
Plus maze used for emergence, exploration, habitat choice and response to bird predator tests. R = refuge for initiating emergence test; A1–A4 = arms to record exploration and for placing of different habitats; PT location of model bird predator.

**Figure 2 animals-14-01315-f002:**
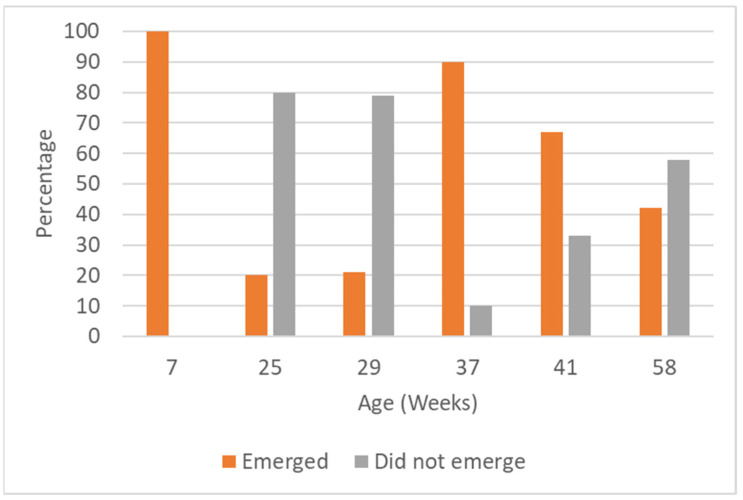
Percentage of trout emerging and not emerging in the emergence test at six different ages.

**Figure 3 animals-14-01315-f003:**
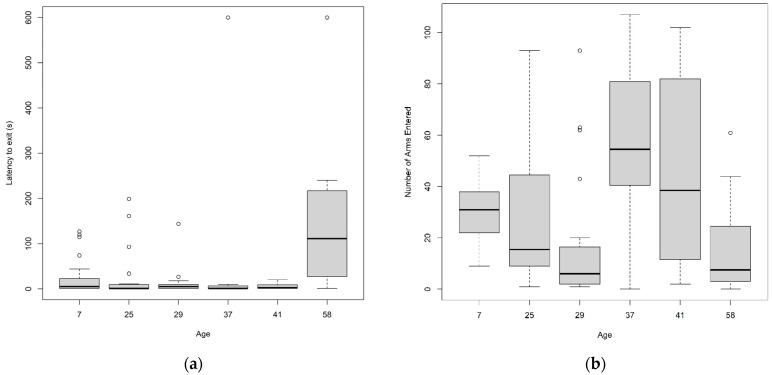
(**a**) Latency of trout at each age to begin exploration in the exploration test and (**b**) number of arms the trout entered at six different ages.

**Figure 4 animals-14-01315-f004:**
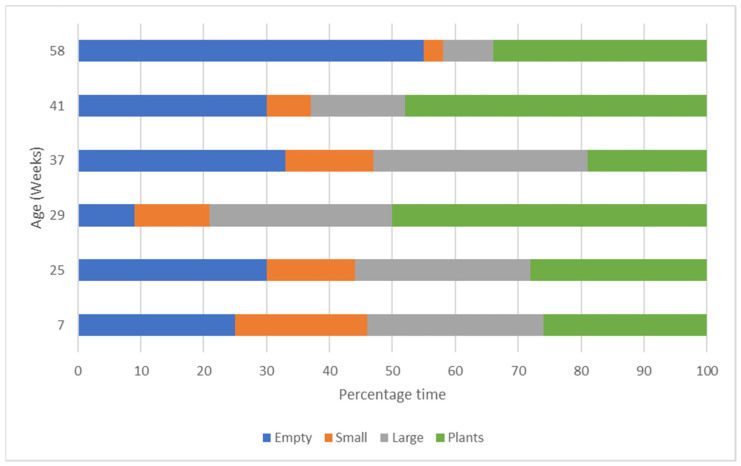
Cluster bar graph of the percentage of time spent in each habitat at six different ages.

**Figure 5 animals-14-01315-f005:**
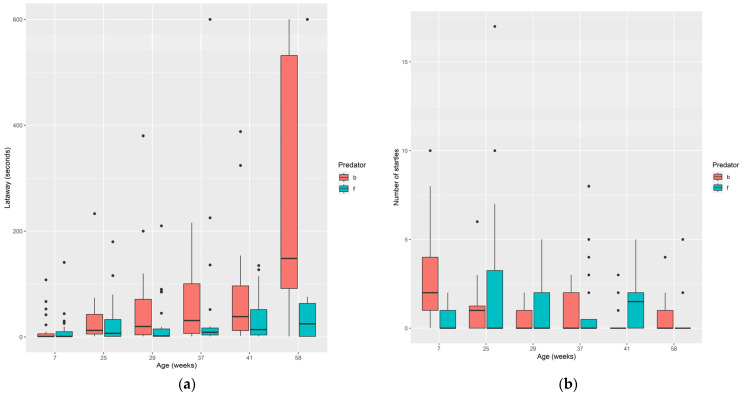
(**a**) Latency (seconds) for fish to move away from bird and fish predators and (**b**) mean number of startle responses at six ages. Summary data for bird (red bars) and fish (blue bars) is presented.

**Table 1 animals-14-01315-t001:** Age, number and size of fish that undertook behavioural tests and were examined for parasites.

Age (Weeks)	N Fish Tested	Mean Length (mm)	SE
7	25	32.4	0.7
25	20	104.8	4.4
29	24	112.9	3.5
37	20	188.9	5.9
41	24	190.3	8.5
58	12	237.0	5.4

**Table 2 animals-14-01315-t002:** Results of Chi-Squared post-hoc tests indicating emergence percentages significantly different from chance.

Age (Weeks)	Residual (χ^2^)	*p* Value
7	4.72	<0.0001
25	−3.80	0.002
29	−4.15	0.003
37	3.12	0.02
41	0.91	1.0
58	−1.23	1.0

## Data Availability

The raw data supporting the conclusions of this article will be made available by the authors on request.

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
