# Peer review of "Age-Related Changes in Survival Behaviour in Parasite-Free Hatchery-Reared Rainbow Trout (Oncorhynchus mykiss)"

_animals, 2024, doi:10.3390/ani14091315_

Round 1

Reviewer 1 Report

Comments and Suggestions for Authors

This paper seeks to investigate the role age-related changes in behavior on a variety of personality/individual differences tests in order to inform on hatchery related management for the breeding of rainbow trout and subsequent release of rainbow trout to bolster stock numbers, as currently many are dying. 

The introduction outlines the environmental and ecological issues well, but there is little to no discussion of the types of tests that are then conducted in the methodology. These tests largely inform on the bold-shy spectrum, but there is no discussion of why the authors think knowledge of this is important to the timing of release of hatred raised fish in the introduction. What is the basis for pursuing this research question? Some discussion of this is needed. 

For each test, what criteria was used to determine bold/shyness? For example, the emergence test investigated latency to emerge from the refuge, how was the time to emerge determined to indicate bold vs shy? These parameters are not clear in the methods or results for what constituted a “longer” vs “shorter” time. Same for the other behavioral tests results, what were the averages observed and what indicated higher or lower than average? Similarly did all fish follow the same order for experiencing each of the tests, or was the test order per subject randomized? If not randomized, what was the rational for this? The time between when tests were conducted is also not clear. May fish species can experience an accumulation of stress from repeated testing that could impact results, if the time frame between tests is not clear or does not allow for a return to baseline for the fish. If there was not a lot fo time to allow a fish to return to baseline, that needs to be discussed and accounted for in the interpretation of the results. In fish dissections, they mention fish were euthanized at the end of all behavioral tests - but the phrasing in other parts of the methods was unclear on if a fish underwent multiple tests or did a given fish only undergo one behavioral test before euthanasia? Some clarification here is needed. Also how many fish underwent each of the given behavioral tests? They have a table for total fish by age, but it is unclear if the distribution of fish that were behaviorally tested is represented by that or if fish did only do one test, the n value per test would be smaller. 

The results appear interesting and are discussed well, but more information is needed to better understand the scope of what these test results are indicating and how they are being interpreted, so they can be effectively used to inform on hatchery breeding management. 

Reviewer 2 Report

Comments and Suggestions for Authors

The authors have exerted significant efforts to carry out this highly intriguing investigation. The writing was remarkable and easily comprehensible. The methods are clearly and explicitly presented. Nevertheless, I have numerous issues regarding the study's design and the output.

In the abstract section, specifically on line 23, the authors stated, "We conclude that knowledge of the behaviour of hatchery reared fish at different ages is useful for decisions around the timing of release…..” Given the extensive and meticulous investigation conducted, this result reached by the researchers is not plausible. This information was already known before conducting the research. The authors should explicitly state the implications of the outcome in a more precise manner.

The introduction portion is quite extensive, yet it may be somewhat elaborate. It can be substantially reduced without altering the intended information it aimed to convey. There is excessive discussion on parasites, often without taking the context into account. The presence of parasites is primarily associated with inadequate hatchery management, and this investigation was carried out under nearly ideal conditions. Hence, the findings of the parasite study do not accurately reflect the actual situation. The authors neglected to take into account water current during the emergence test. The fish were placed in a 14 by 18 centimeter enclosure. The authors' decision to introduce fish measuring approximately 18.8 to 23.7 cm (at 37 to 58 weeks of age) into the study, with the expectation that these fish would exhibit natural behavior, is truly perplexing. The authors did not take into account water velocity or visibility when conducting the substrate test, predator test, and novel food test. In my opinion, the dimensions of the test tanks used in the substrate test were insufficient to accommodate the larger fish and hindered their ability to display their innate behavioral patterns. The duration of acclimatization, which was set at 5 minutes, proved to be insufficient. In my opinion, it is more crucial to document the initial reaction of predators rather than spending 10 minutes on verification. Why did the authors conduct a 10-minute examination of it? Did the authors take into account the particle size when conducting the novel food test? Why was the same bait used for fish of all sizes and ages?

If the authors performed all the experiments in succession, there is a strong likelihood that they were under significant stress, which could have hindered their ability to display natural behavior. Based on the comprehensive investigation, it was found that fish between 37-41 weeks old exhibit more bold behavior, while fish around 58 weeks old tend to be somewhat shy. In my opinion, their study has not adequately fulfilled its objectives as stated in the introduction, which were to assess the likelihood of fish survival and its impact on the ecosystem. 

I believe there are fundamental errors in the design of the experiment. Despite finding the matter intriguing, I must reject the manuscript. I extend my sincerest wishes for success and prosperity to the authors.

Reviewer 3 Report

Comments and Suggestions for Authors

1. The research topic is rather important because the leading cause of mortality in hatchery-reared juvenile fish in the wild is due to deficits in behavioral development resulting from captive rearing conditions.

The introduction provides sufficient background and include all relevant reference.

2. In the introduction, the authors predicted that in response to prolonged domestication 1) older fish would be bolder as revealed by taking more risks with predators and more readily taking novel foods. 2) They also predicted that older fish would have more parasites than younger fish.

Thus, the study was expected to consist of two rather disparate parts: 1) behavioral experiments and 2) studies of parasites.

The results are also expected to consist of two general parts:

3.1. Experiments (1, 2, 3, 4) and

3.2. Parasitic infections study

It is proposed to reorganize the results to make them more logical and consistent with the stated goals.

In Materials and methods, rename paragraph 2.5. “Fish dissections” to “Study of parasites”, or something like that.

3. In the experimental part, the authors tested the behavior of hatchery-reared rainbow trout at six different ages to assess their chances of survival and impact on the ecosystem.

The research design corresponds to the stated objectives. The special design of the test tanks is shown. Experimental methods and procedures are adequately described.  The average length of the fish is noted. The acclimatization period (which is important) was used in the experiment. Statistical analysis is objective.

My suggestion for future studies of this type is to use more fish in each set of experiments. The number of fish used here in some sets (12, 20) is on the permissible limit.

4. The results are presented quite clearly.

But I suggest using more illustrations in the text (at least two of them should be moved from the Appendix to the article itself):

Figure S2: Photo of Nankeen Night Heron (Nycticorax caledonicus) used for the avian predator avoidance test. [This curious  photo will make the article more attractive to readers].

Figure S6:  Gill arch with filaments of 29-week-old trout. (but not “Gill filaments and operculum”). [This will make evidence that your fishes were found to be in good health].

5.  I did not find any explanation in the article as to whether the Labyrinth Tank Plus is an original design or has been used previously in similar experiments. Please add this information.

The manuscript may be accepted with minor changes.

Author Response

See attchment

Round 2

Reviewer 1 Report

Comments and Suggestions for Authors

I found the authors response and changes to the manuscript adequately addressed the comments and questions I had raised and provide clarification and context where it was needed.

Reviewer 2 Report

Comments and Suggestions for Authors

Thank you for submitting your revised version. As you could see and agree on some occasions that there were a few flaws in the design of the study which are not possible to correct. I am sympathetic to the efforts the authors have given to conduct the study, and thereby I am giving my recommendations to the editors.